# Epicardial Adiposity in Relation to Metabolic Abnormality, Circulating Adipocyte FABP, and Preserved Ejection Fraction Heart Failure

**DOI:** 10.3390/diagnostics11030397

**Published:** 2021-02-26

**Authors:** Jiun-Lu Lin, Kuo-Tzu Sung, Yau-Huei Lai, Chih-Hsuan Yen, Chun-Ho Yun, Cheng-Huang Su, Jen-Yuan Kuo, Chia-Yuan Liu, Chen-Yen Chien, Ricardo C. Cury, Hiram G. Bezerra, Chung-Lieh Hung

**Affiliations:** 1Division of Endocrinology and Metabolism, Department of Internal Medicine, MacKay Memorial Hospital, Taipei 10449, Taiwan; lululin.6454@mmh.org.tw; 2Graduate Institute of Clinical Medicine, College of Medicine, National Taiwan University, Taipei 10617, Taiwan; 3Department of Medicine, Mackay Medical College, New Taipei City 25245, Taiwan; chsu007@gmail.com (C.-H.S.); liu.chiayuan@gmail.com (C.-Y.L.); 4Division of Cardiology, Department of Internal Medicine, MacKay Memorial Hospital, Taipei 10449, Taiwan; 8905012@gmail.com (K.-T.S.); yuanjim.5817@mmh.org.tw (C.-H.Y.); jotaro3791@gmail.com (C.-L.H.); 5Division of Cardiology, Department of Internal Medicine, MacKay Memorial Hospital, Hsinchu 30071, Taiwan; garak1109@mmh.org.tw; 6Mackay Junior College of Medicine, Nursing, and Management, Taipei 11260, Taiwan; med202657@gmail.com; 7Department of Radiology, MacKay Memorial Hospital, Taipei 10449, Taiwan; 8Institute of Biomedical Sciences, Mackay Medical College, New Taipei City 25245, Taiwan; 9Division of Gastroenterology, Department of Internal Medicine, MacKay Memorial Hospital, Taipei 10449, Taiwan; 10Cardiovascular MRI and CT Program, Baptist Cardiac Vascular Institute, Miami, FL 33176, USA; ricardocury@hotmail.com; 11Cardiovascular Department, University Hospitals Case Medical Center, Cleveland, OH 44106, USA; hiram.bezerra@uhhospitals.org

**Keywords:** epicardial adipose tissue (EAT), pro-inflammatory cytokines, metabolic syndrome, adipocyte fatty acid-binding protein (A-FABP), heart failure (HF), preserved ejection fraction heart failure (HFpEF), strain, strain rate

## Abstract

Epicardial adipose tissue (EAT) as a source of pro-inflammatory cytokines tightly linked to metabolic abnormalities. Data regarding the associations of EAT with adipocyte fatty acid-binding protein (A-FABP), a cytokine implicated in the cardiometabolic syndrome, might play an important part in mediating the association between EAT and cardiac structure/function in preserved ejection fraction heart failure (HFpEF). We conducted a prospective cohort study comprising 252 prospectively enrolled study participants classified as healthy (*n* = 40), high-risk (*n* = 161), or HFpEF (*n* = 51). EAT was assessed using echocardiography and compared between the three groups and related to A-FABP, cardiac structural/functional assessment utilizing myocardial deformations (strain/strain rates) and HF outcomes. EAT thickness was highest in participants with HFpEF (9.7 ± 1.7 mm) and those at high-risk (8.2 ± 1.5 mm) and lowest in healthy controls (6.4 ± 1.9 mm, *p* < 0.001). Higher EAT correlated with the presence of cardiometabolic syndrome, diabetes and renal insufficiency independent of BMI and waist circumference (*p*_interaction_ for all > 0.1), and was associated with reduced LV global longitudinal strain (GLS) and LV mass-independent systolic/diastolic strain rates (SRs/SRe) (all *p* < 0.05). Higher A-FABP levels were associated with greater EAT thickness (*p*_interaction_ > 0.1). Importantly, in the combined control cohort, A-FABP levels mediated the association between EAT and new onset HF. Excessive EAT is independently associated with the metabolic syndrome, renal insufficiency, and higher A-FABP levels. The association between EAT and new onset HF is mediated by A-FABP, suggesting a metabolic link between EAT and HF.

## 1. Introduction

Owing to excessive adiposity, obesity is an important key clinical risk in the pathophysiology of heart failure with preserved ejection fraction (HFpEF), an emerging HF phenotype worldwide in both Western and Asian populations [1,2]. Conventional use of body mass index (BMI) fails to differentiate true body fat fractionation and therefore is neither specific nor accurate as an ideal marker for quantifying the total burden of adiposity [3]. Accumulating data proposed that excessive visceral adiposity, rather than BMI, is a central pathological player mediating metabolic derangements and may serve as a major source of pro-inflammatory cytokines, leading to cardiovascular disorders [3,4,5]. Epicardial adipose tissue (EAT), as visceral adiposity confined within the pericardial sac that tightly regulates myocardial structural and functional homeostasis [6], has recently emerged as an attractive research topic in the pathogenetic mechanisms of HFpEF.

Recently, the Asia-Pacific region has been facing a metabolic syndrome (MetS) epidemic [7]. It is now evident that patients with HFpEF have a high prevalence of multimorbidity tightly associated with MetS components [1,8]. In particular, a large, prospectively enrolled multi-ethnic Asian HFpEF registry further illustrated multi-metabolic abnormality as a geography-specific Asian HFpEF phenotype [9]. A-FABP (also known as FABP4 or aP2) is a member of the intracellular FABP family predominantly expressed in mature adipocytes, and it has been shown to display higher expression levels in human EAT and central aortic vasculature in overweight/obese subjects or those with severe MetS [10] and associated with unfavorable LV remodeling in obese women [11]. Moreover, emerging evidence reveals that adipocyte fatty acid-binding protein (A-FABP), a circulating marker enriched in and released from visceral adipose tissue, increases correspondingly with incident MetS [12] and may suppress cardiomyocyte contractility and contribute to HF development [13]. While EAT has been demonstrated to provoke a systemic inflammatory cascade [14] as the main hypothetical pathophysiology in the development of HFpEF, the possible mediating role of A-FABP between EAT, cardiac structure, and function, along with its correlation with HFpEF, has never been thoroughly explored. To investigate whether EAT may distribute differently in concert with the degree of cardiometabolic myocardial dysfunction, we aimed to compare EAT across a broad spectrum of cardiovascular disease (CVD) categories comprising healthy and high metabolic risk subjects.

## 2. Methods

### 2.1. Study Subjects

From December 2011 to September 2014, patients from the outpatient clinics of a tertiary medical center (Mackay Memorial Hospital, Taipei, Taiwan) were prospectively enrolled. All study participants gave written informed consent, and the institutional review board at Mackay Memorial Hospital approved the study (11MMHIS127; 15MMHIS031e) (21 March 2015). All research was performed in accordance with relevant guidelines/regulations. The primary goal of this study was to explore the clinical significance and relevance of epicardial adiposity with circulating pro-inflammatory (hs-CRP) and HF biomarkers (plasma B-type natriuretic peptide (BNP), galectin-3, PIIINP, and A-FABP) and adverse cardiovascular endpoints across a broader spectrum of cardiovascular including those with known cardiometabolic risk factors and HFpEF, defined by subjects with prior admission for HF with an LV ejection fraction of ≥50%. According to the Adult Treatment Panel (ATP) III and modified Taiwanese guideline by ethnic Asian population, cardiometabolic risk was defined as the presence of any of the following conditions: (1) Central obesity: Waist circumference ≥90 cm in men or ≥80 cm for women; (2) abnormal blood pressure: Systolic blood pressure ≥130 mmHg, diastolic blood pressure ≥85 mmHg, or history of diagnosed hypertension; (3) abnormally high triglycerides ≥150 mg/dL; (4) low high-density lipoprotein (HDL) <40 in men and <50 mg/dL in women; (5) dysglycemia: Fasting plasma glucose ≥100 mg/dL or prior known diabetes history. Subjects with at least three cardiometabolic risk factors were defined as having metabolic syndrome (MetS). Patients with atrial fibrillation, moderate-to-severe valvular heart disease, or prior hospitalization for systolic heart failure were excluded (*n* = 9). The study flowchart is illustrated as Appendix A.

Original study participants included three groups: (1) Healthy controls (*n* = 40) without known cardiovascular risk factors or systemic diseases; (2) high-risk (*n* = 161), those who exhibited ≥1 cardiometabolic component using the Adult Treatment Panel III and modified guideline for ethnic Asians [15]; and (3) HFpEF (*n* = 51).

In this study, a history of hypertension was defined as systolic blood pressure >140 mmHg, diastolic blood pressure >90 mmHg, or previously diagnosed hypertension under pharmaceutical control. Diabetes mellitus (DM) was defined as a fasting glucose level >126 mg/dL or previously diagnosed DM under pharmaceutical control. Dyslipidemia medication history was defined as current usage of lipid-lowering drugs of any kind, such as statins or fibrates. CVD was defined as prior history of myocardial infarction, coronary artery post angioplasty, or history of cerebrovascular events. The study setting and design have been published previously [16].

### 2.2. Anthropometric Measurements

All baseline characteristics and information regarding anthropometric measures were collected, including age, height, weight, waist circumference, and blood pressure. BMI was calculated as weight (kg) divided by the square of the body height (m) (i.e., kg/m^2^). To measure the waist circumference (in centimeters), the waist-tape was placed horizontally around the midpoint between the lower rib margin and iliac crest. We used waist circumference cut-offs of 80 cm in women and 90 cm in men as the threshold of abnormal central obesity reflecting excessive abdominal fat accumulation [5]. A standardized cuff sphygmomanometer was used to obtain resting blood pressures, which were measured by medical staff members blinded to other clinical information or laboratory test results.

### 2.3. Biochemical Analysis of Pro-Inflammatory and HF Markers

A Hitachi 7170 Automatic Analyzer (Hitachi Corporation, Hitachinaka Ibaraki, Japan) was used to measure levels of fasting glucose (hexokinase method), creatinine (kinetic colorimetric assay), total cholesterol and triglyceride, and alanine aminotransferase (enzymatic method). Lipid profiles including low-density and high-density lipoprotein-cholesterol were obtained using homogenous enzymatic colorimetric assay. High-sensitivity C-reactive protein (hs-CRP) levels were determined using a highly sensitive, latex particle-enhanced immunoassay (Elecsys 2010; Roche Diagnostics GmbH, Mannheim, Germany). BNP (Biosite Inc., Alere, France), galectin-3 (R&D Systems, Minneapolis, MN, USA), N-terminal pro-peptide of type III procollagen (PIIINP) (Orion Diagnostics, Fountain Hills, AZ, USA) and adipocyte FABP (A-FABP) (BioVendor, Inc., Brno, Czech Republic) concentrations were determined using commercially available enzyme-linked immunosorbent assay kits.

### 2.4. Measures of EAT, and Cardiac Structure and Function

Each subject underwent two-dimensional (2D) and M-mode transthoracic echocardiography using Vivid 7 (GE Vingmed Ultrasound, Horten, Norway) equipped with a 2.5-MHz to 4.5-MHz transducer, with images stored as Digital Imaging and Communications in Medicine format. Standard parasternal and apical views were obtained in the left lateral decubitus position. EAT is generally identified as the echocardiographic free space between the outer wall of the myocardium and the visceral layer of the pericardium, and its thickness was obtained at end-systole (Appendix A). The comparison and validation of echocardiography-derived EAT with multiplanar reconstructions of multi-detector computed tomography (CT) data was performed by Lai et al [17] and showed excellent correlations in our imaging lab (*n* = 178, including comparisons of EAT with corresponding planes using contrast CT (*n* = 58) and 120 non-contrast CT, R^2^ = 0.78 and 0.79 for long- and short-axis data) with good reproducibility (intra-observer/inter-observer coefficients of variation (COV) for long-axis and short-axis EAT: 5.4% and 6.2%; 5.8% and 6.9%, respectively).

LV end-diastolic and systolic diameters, LV posterior wall thickness, interventricular septum thickness, peak early diastolic mitral flow velocity (E), peak late diastolic mitral flow velocity (A), E/A ratio, and deceleration time of early diastolic mitral flow were all obtained according to the American Society of Echocardiography guidelines. LV mass index was calculated from the LV end-diastolic diameter and septal and LV posterior wall thickness using the validated Devereux formula. Tissue-Doppler imaging (TDI) determined peak myocardial systolic (TDI-s’) and early diastolic relaxation velocity (TDI-e’) were determined from septal and lateral basal myocardial segments. Myocardial deformational indices were measured using novel offline proprietary analysis system (EchoPAC PC, Version 110.0.2; GE Medical Systems, Horten, Norway) from baseline 2D images obtained from three short-axis views (including mitral, papillary muscle, and apical levels) for LV circumferential strain; three LV apical views (including 2-chamber, 4-chamber, and 3-chamber views) for longitudinal LV strain and strain rate components including systolic (SRs), early (SRe), and late diastolic (SRa), with twist analysis quantified by subtracting rotation from LV mitral annulus (minus in data presentation) to LV apical level (positive in data presentation) as net angle differences as detailed in our previous work [18]. Representative global LV longitudinal (GLS) and circumferential (GCS) strain, together with strain rate components including SRs, SRe, and SRa, were averaged from three LV apical views in each study participant.

### 2.5. Validating EAT with CT-Based PCF Measurement

Among 45 subjects with paired CT-based three-dimensional (3D) construction for PCF (16-slice multi-detector CT scanner, Sensation 16; Siemens Medical Solutions, Forchheim, Germany), using novel offline proprietary software (Aquarius 3D Workstation, TeraRecon, San Mateo, CA, USA) and EAT available (Appendix A), we showed fair correlation between EAT and 3D CT-based PCF (*r* = 0.73, +19.5 [95% confidence interval: 13.9–25.1] ml PCT per 1 mm increment of EAT) [19].

### 2.6. Statistical Analysis

Data for continuous variables are expressed as mean ± standard deviation (SD) and categorical variables as frequencies and proportions of occurrence. Differences of baseline demographics and anthropometric and cardiometabolic parameters among the three groups were tested using analysis of variance (ANOVA), with categorical data analyzed using the χ^2^ test or Fisher’s exact tests as appropriate. Post-hoc comparisons between each group were further performed using the Bonferroni multiple-comparison test. The relationship among EAT, echo-derived parameters, and pro-inflammatory/HF biomarkers were determined using Pearson’s correlation analysis. A multivariate logistic regression model was used to determine the significance of covariate-adjusted relations between epicardial fat, clinical co-morbidities, and HFpEF with individual odds ratios, p values, and 95% confidence intervals. From our previous work, EAT based on echocardiographic measurements in subjects with higher cardiovascular risk or MetS were similar to those used for the currently defined “high-risk group” and had a mean value of 8.2 mm (SD: 1.0). In addition, Parisi et al. presented a EAT mean value of 8.6 mm (SD: 2.55) in subjects with systolic HF [20]. While HFpEF subjects are more likely to be obese and predominantly elderly women, we expected a mean EAT of nearly 8.5–9.5 mm for HFpEF in the present study. To identify an EAT mean difference of nearly 1 mm with a SD of 2.4–2.5 mm (effect size: 0.4 by [Cohen’s d]) with greater than 90% power and an α error (*p* value) of 0.05 based on a 3:1 proportion between “high-risk” and HFpEF individuals, a sample size of 150:50 was required for study participants. For associations with EAT as outcome measures, we explored anthropometric and clinical determinants of EAT by using backward stepwise regression models, with systolic blood pressure and diastolic blood pressure entered separately due to co-linearity. By using multi-variate linear regression models adjusted for age, sex, body mass, and LV mass in multivariable models, we further examined the mechanistic effects of greater EAT on a variety of echocardiography-defined cardiac structural and functional measures including TDI and myocardial speckle-tracking parameters.

In the present study, mediator analysis was constructed to identify a potential intermediary role of several key cardiometabolic pro-inflammatory/HF biomarkers (e.g., hs-CRP, BNP, gelactin-3, PIIINP, and A-FABP) in mediating the association of EAT with clinical HFpEF by constructing a causal model based on the hypothesis that HF severity may be partly modulated by EAT via these circulating markers. Bootstrapping analysis was also applied to ensure the role of these biomarkers as potential mediators.

The p values were two-tailed, with *p* < 0.05 considered statistically significant. All statistical analyses were performed using Stata version 12.0 (StataCorp LP, College Station, TX, USA).

## 3. Results

### 3.1. Clinical Demographical and Metabolic Relevance of EAT

Clinical demographic and echocardiographic characteristics of the study groups at enrollment are summarized in Appendix A. Among 264 study participants, 252 (mean age: 65.8 ± 9.9 years; 64.7% female) met our final inclusion/exclusion criteria for comprehensive echocardiography analysis (Appendix A). EAT measurement was highest in participants with HFpEF (9.7 ± 1.7 mm), higher in high-risk participants (8.2 ± 1.5 mm) and lowest in healthy controls (6.4 ± 1.9 mm, *p* < 0.001). By categorizing study participants into EAT tertiles, those with greater EAT in tertile groups had more advanced age; were more likely to be female; have higher blood pressure, greater waist circumference, body mass index (BMI), body fat, fasting glucose, uric acid, and triglyceride; and lower HDL-C and poor renal function in terms of a lower eGFR (all trend *p*: <0.05) (Table 1). Abnormal phenotypic obesity, either defined by abnormal BMI (>27.5 kg/m^2^) or waist circumference (sex-specified cut-offs), had a consistently higher prevalence of metabolic syndrome (MetS) across EAT tertiles (*p* < 0.05). Notably, subjects in the second EAT tertile group (7.6–9.0 mm) without phenotypic obesity almost doubled and tripled the risk of having MetS compared to first EAT tertile despite having a lower BMI (60% vs. 31%) and normal waist circumference (28% vs. 9%), respectively. More advanced age, female sex, greater BMI, higher blood pressure, lower HDL-C, poor renal function, presence of diabetes, and presence of HFpEF all were independent determinants for higher EAT (all *p* < 0.05) in multivariate regression models (with systolic blood pressure and diastolic blood pressure into models separately) (Table 2). Significant associations (*p* < 0.005) were found between abundant EAT (per 1 mm increment) with HFpEF (OR: 1.61, 95% CI: 1.22 to 2.12), chronic kidney disease (OR: 1.51, 95% CI: 1.16 to 1.96), and diabetes (OR: 1.48, 95% CI: 1.17 to 1.87), but not CVD (OR: 1.23, 95% CI: 0.92 to 1.66; *p* = 0.15) or hypertension (*p* = 0.15) (Appendix A).

### 3.2. Associations of EAT with Pro-Inflammatory/HF Markers

Higher EAT was associated with higher hs-CRP, galectin-3, PIIINP, and A-FABP and marginally higher BNP (all trend *p* < 0.05) (Appendix A). Among five pro-inflammatory and HF biomarkers, EAT showed positive linear associations with hs-CRP (*r* = 0.26), BNP (*r* = 21), gelactin-3 (*r* = 0.26), PIIINP (*r* = 0.37), and A-FABP (*r* = 0.41) (all *p*
*<* 0.001). A significant, graded increase of A-FABP across quartiles of EAT was observed (*p* for trend < 0.001). Based on multivariate regression, increased EAT (per 1-mm EAT increment) was an independent predictor for higher A-FABP level, but not other pro-inflammatory and HF markers; the trend persisted after adjustment for age, sex, BMI, total cholesterol, high-density lipoprotein cholesterol, and past medical history including hypertension, CVD, DM, and HFpEF (Coefficient: 2.00, 95% CI: 0.44 to 3.53, *p* = 0.012).

### 3.3. Associations of EAT with Cardiac Structure and Function

Subjects with greater EAT are also more likely to have co-morbid hypertension, diabetes, known cardiovascular diseases, and HFpEF (all *p* < 0.05) (Table 1); in addition, they exhibited larger ventricular wall thickness, LV mass, and reversed E/A ratio and markedly lower myocardial relaxation e’, attenuated myocardial systolic velocity s’, higher E/e’, global ventricular systolic function (GLS), and LV myocardial systolic/diastolic strain rates (SRs/SRe) (all trend *p*: <0.05) (Table 1); global LV ejection fraction (LVEF) and global circumferential strain (GCS) were relatively unchanged. In general, greater EAT was associated with more unfavorable LV structural remodeling, including greater wall thickness, larger LA volume, lower E/A ratio, poor myocardial TDI-s’/TDI-e’, higher LV filling E/e’, and poor systolic myocardial deformations GLS and LV SRs/SRe (all linear p trend *p* < 0.05) and with a slightly increased SRa (Table 1). Further adjustment for clinical variables did not attenuate the significance level between greater EAT burden with poor LV GLS and reduced LV strain rates SRs and SRe (Figure 1, all *p* < 0.05).

### 3.4. Association of EAT with Incident HF: Mediator Analysis

During follow-up (median: 3.8 years, IQR: 3.5–4.6 years), 63 of 252 study participants had hospitalization for HF. Increased EAT (per 1 mm EAT increment) was associated with higher HF events (Table 3) (adjusted HR: 1.36 [1.14–1.64], *p* = 0.001) and composite HF/death events (adjusted HR: 1.35 [1.14–1.60], *p* = 0.001) based on multivariate Cox regression analysis. Using EAT and A-FABP from the receiver operating characteristic analysis for identifying the baseline presence of HFpEF yielded optimal clinical cut-offs of 8.7 mm and 24.8 ng/mL for EAT and HFpEF, respectively. Those classified with abnormally high EAT (≥8.7 mm) experienced seven-fold more HF events (HR: 7.79 [95% CI: 3.94 to 15.43]), *p* < 0.001. Adding FABP strata (<, ≥24.8 ng/mL) further successfully discriminated HF re-hospitalization from the original EAT cut-off (<, ≥8.7 mm) in the multivariate Cox models (Figure 2, adjusted HR: 12.7, [95% CI: 1.8 to 96.0], adjusted HR: 31.3 [95% CI: 4.3 to 228.8] for EAT ≥ 8.7 mm/FABP < 24.8 ng/mL and EAT ≥ 8.7 mm/FABP ≥ 24.8 ng/mL, respectively, using EAT < 8.7 mm as a reference value).

Mediator analysis was further constructed to examine potential effects of several key pro-inflammatory/HF markers in the association of EAT with HF endpoint, under the hypothesis that these markers can play biological intermediary roles in mediating EAT and HF outcomes. After adjustment for age, sex, and BMI, the mediator analysis showed that BNP, PIIINP, and FABP appear to be active markers mediating effects in the relationship of EAT with incident HF. After a more detailed adjustment for baseline medical histories, FABP was found to be the only mediator, explaining 8.95% mediating effects in the relationship between EAT and incident HF.

## 4. Discussion

This study defines several key ideas regarding the clinical presentation, associated comorbidities, and cardiac structure/function associated with utilizing echocardiography-based EAT. Firstly, greater EAT thickness was tightly associated with several clinical cardiometabolic factors and was markedly greater in subjects with type 2 DM, HFpEF, and renal insufficiency, independent of BMI. Secondly, excessive EAT was positively correlated with unfavorable cardiac remodeling and was inversely correlated with diastolic and subclinical systolic function. Third, greater EAT was associated with higher circulating pro-inflammatory/HF markers including hs-CRP, galectin-3, BNP, and PIIINP and was independently associated with A-FABP. Fourth, an EAT cut-off of 8.7 mm was independently associated with higher HF events, with higher EAT accompanied by DM or elevated FABP demonstrating worse outcomes.

### 4.1. Functional and Prognostic Significance of EAT as a Surrogate of Visceral Obesity

EAT, as part of the pericardial fat, features anatomical and functional contiguity to epicardial coronary arteries and the myocardium [21]. Echocardiography defined EAT may be better characterized by MRI-defined visceral fat than the conventional surrogates of central obesity as a key pathophysiological determinant for metabolic syndrome (MetS), insulin resistance, and type 2 diabetes [22,23,24]. As a greater EAT is associated with diastolic dysfunction in morbid obesity and DM [25,26], we demonstrated that greater EAT was independent predictor systolic/diastolic mechanical myocardial indices using deformations (e.g., GLS, LV SRs, and SRe) beyond BMI information, indicating negative effects of excessive EAT on myocardial contractile mechanics despite preserved LVEF. Except for its relevance with diastolic dysfunction, HF with LVEF >40% reportedly had larger EAT than normal controls despite having similar BMI [27]; conversely, HFrEF patients appeared to have diminished EAT [28].

Due to the anatomical and functional contiguity to the myocardium, EAT may locally affect the heart to a greater extent when compared to visceral adiposity from other regions [6]. For example, the possible physical constriction effect limiting myocardial relaxation during the diastolic phase caused by EAT [25], local paracrine signaling/cytokines from oxidative stress and lipotoxicity (FFA hyper-influx) [29] in a dysglycemic status, together with elicited downstream pro-inflammatory cascades, have all been proposed as part of the pathophysiology of “visceral adiposity syndrome” [5]. Additional mechanisms by which excessive EAT may contribute to HFpEF pathogenesis include its deleterious effects on microvascular/endothelial function accompanied by diverse metabolic disorders and systemic pro-inflammatory signaling, leading to excessive extracellular matrix turnover/fibrosis, structural remodeling, impaired cardiac lusitropic/contractile properties with increased myocardial stiffness predisposing to HFpEF, and co-morbid renovascular dysfunction [30,31,32,33]. Herein, we demonstrated that larger EAT measure may confer greater cardiovascular events for HF (HR: 7.43, 95% CI: 3.87 to 14.26, *p* < 0.001) with a cut-off of 8.7 mm. To our knowledge, this is the first study to display evidence that echocardiography-based EAT assessment is a surrogate of visceral adiposity for predicting HF.

### 4.2. Associations of EAT with Circulating A-FABP

As aforementioned, elicited myocardial fibrosis and excessive extracellular matrix turnover/degradation through chronic pro-inflammatory signaling are considered major pathophysiological culprits from diastolic dysfunction to HFpEF [14,31,32]. Indeed, greater EAT showed positive correlations with several circulating HF markers, reflecting up-regulated pro-inflammatory or extracellular matrix turnover/degradation; this has demonstrable prognostic implications, including hs-CRP, gelactin-3, PIIINP, and A-FABP [34,35,36,37]. Local paracrine signaling mediates cardiomyocyte dysfunction through A-FABP by adjacent EAT in a dose-dependent manner via intracellular Ca^2+^ regulation in experimental models [13], supporting its inhibitory role in cardiomyocyte excitation–contraction. High plasma FABP4 levels as a predictor of HF development has also been shown in a large-scale prospective study with 10.7 years of follow-up [38].

Consistent with the above finding, we demonstrated an association between greater EAT and higher FABP independent of BMI, though FABP was not associated with DM when BMI was considered. Given these associations, we speculated that A-FABP correlates with greater BMI and may directly reflect the burden of visceral adiposity (e.g., EAT) as a marker of excessive EAT, and higher A-FABP likely plays an adjunctive pathophysiological role in mediating EAT-related “cardiomyopathy” (Figure 3). Herein, we further demonstrated that A-FABP was the most predominant mediator in the association of EAT with HFpEF endpoint with a specific threshold proposed (A-FABP more or less than 24.8 ng/mL). Our findings highlight the pathophysiological intermediary role of A-FABP in the molecular signaling linking EAT, development of HFpEF, and further adverse clinical outcomes.

### 4.3. Limitations

Despite our comprehensive analysis of cardiac structure, function, and several pro-inflammatory or HF markers with EAT expansion, the sample size in this cohort observational study was relatively small; thus, a larger population may help replicate our findings. As part of the total body visceral adiposity measure, the use of EAT as a surrogate in determining the relations with circulating biomarkers may not accurately characterize the true biological influence of visceral fat in these associations. Secondly, quantification of EAT using echocardiography method can be challenging and may not accurately reflect total epicardial fat burden in certain situations (e.g., in obese subjects) when compared to MRI or CT measures. In fact, visceral adiposity can be highly heterogeneous in components and bioactivity [39] over whole body distribution; therefore, the use of EAT as a pathological marker of body visceral adiposity can lead to over-simplified systemic and remote effects from whole body visceral fat burden in any individual. Notably, data comparing EAT with abdominal fat on these outcome measures were not feasible due to the lack of abdominal visceral fat in study design. Besides, more precise myocardial pathological changes at the cellular level, including diffuse or focal fibrosis, underlying cardiac structural and functional changes and HFpEF accompanying excessive EAT were not assessed in this study. Further detailed tissue disarrangements (e.g., either by upregulated extracellular fibrotic deposition/turnover or scar formation) detected using cardiac magnetic resonance imaging may provide more in-depth insights in future studies. Finally, whether the strong associations of EAT with DM regardless of body mass and circulating A-FABP were unique and limited to ethnic Asians may warrant large-scale, multi-ethnic study designs to validate our findings.

## 5. Conclusions

Prevalent DM, HFpEF, and renal insufficiency in a high cardiometabolic risk patient population was associated with greater EAT, which is tightly linked to adverse ventricular structural remodeling and poor myocardial functions. As a predictor of greater EAT burden, A-FABP, a novel pro-inflammatory mediator closely linked to EAT burden, is capable of further discriminating HF outcomes adjunctive to EAT. To the best of our knowledge, this is the first study to explore the associations of EAT and DM phenotypes in ethnic Asians as well as HF outcomes indicating that EAT serves as key player for HF, rather than BMI, in ethnic Asians.

## Figures and Tables

**Figure 1 diagnostics-11-00397-f001:**
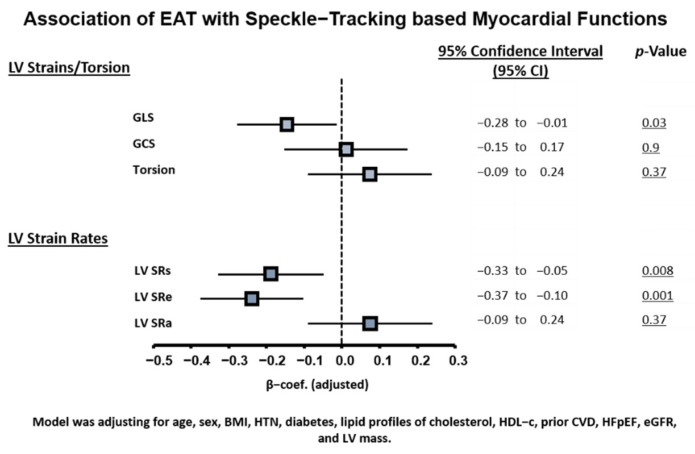
Associations of EAT burden with myocardial mechanics based on speckle-tracking. Model was adjusted for age, sex, BMI, hypertension, diabetes, lipid profile of cholesterol and HDL-C, prior CVD, HFpEF, eGFR, and LV mass. CVD, cardiovascular disease; eGFR, estimated glomerular filtration rate; HDL-C, high density lipoprotein cholesterol; HFpEF, heart failure with preserved ejection fraction; LV, left ventricle; MetS, metabolic syndrome. All strain and strain rate values are reported as absolute values |x| and further standardized.

**Figure 2 diagnostics-11-00397-f002:**
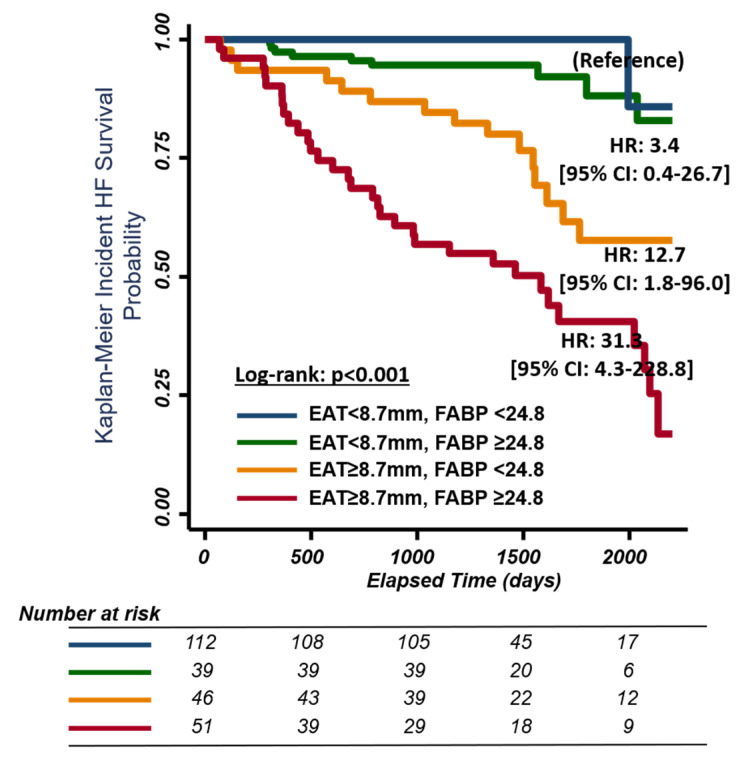
Kaplan–Meier curves for survival free of heart failure events (including heart failure hospitalization or death) stratified by EAT and A-FABP level. EAT, epicardial adipose tissue; FABP, fatty acid-binding protein. EAT cut-off point <8.7 mm and ≥8.7 mm. FABP cut-off point <24.8 mm and ≥24.8 mm.

**Figure 3 diagnostics-11-00397-f003:**
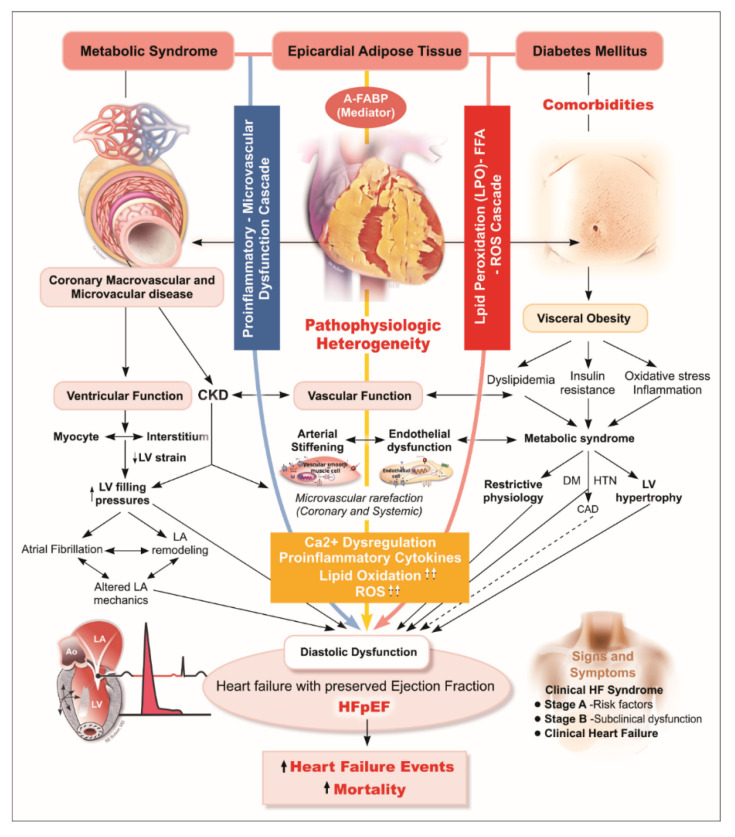
Diagram illustrating hypothetical pathological link between excecssive EAT, A-FABP, and HFpEF. In HFpEF, myocardial remodeling and diastolic dysfunction are driven by endothelial inflammation, a physical constriction effect limiting myocardial relaxation during diastolic phase caused by EAT, and EAT-related paracrine signaling/cytokines from oxidative stress and lipotoxicity (FFA hyper-influx). A-FABP, a main mediator enriched in and released from visceral adipose tissue, is believed to elicit downstream pro-inflammatory cascades, has been proposed as part of the “visceral adiposity syndrome”. Deteriorating microvascular/endothelial function accompanying diverse metabolic syndrome-related comorbidities and systemic pro-inflammatory signaling, leading to excessive extracellular matrix turnover/fibrosis, structural remodeling, impaired cardiac lusitropic/contractile properties with increased myocardial stiffness predisposing to HFpEF, and co-morbid renovascular dysfunction. A-FABP, adipocyte fatty acid-binding protein; HFpEF, heart failure with preserved ejection fraction.

**Table 1 diagnostics-11-00397-t001:** Clinical demographics and echocardiographic measures of participants categorized by EAT tertile groups.

Metabolic Score Categories	All Subjects (*N* = 252)	Epicardial Adipose Tissue (EAT)			*p* for Trend	*R* Value (All Variable Correlated with EAT)	*p* Value (Pearson Correlation)	ANOVA/ꭓ^2^
		Q1 (*n* = 84) ≤7.5 mm	Q2 (*n* = 84) 7.6–9.0 mm	Q3 (*n* = 84) ≥9.1 mm				
Baseline Demographics								
Age, y	65.8 ± 9.87	61.7 ± 7.79	65.8 ± 10.1 *	69.7 ± 10.0 *^,#^	<0.001	0.381	<0.001	<0.001
Female sex, *n* (%)	165 (65.5%)	47 (56.0%)	53 (63.1%)	65 (77.4%)	0.004	0.153	0.02	0.01
Systolic blood pressure, mm Hg	140.3 ± 20.0	133.4 ± 19.1	140.3 ± 19.3 *	147.2 ± 19.3 *	<0.001	0.303	<0.001	<0.001
Diastolic blood pressure, mm Hg	80.8 ± 12.3	78.6 ± 11.6	80.1 ± 12.1	83.6 ± 12.6 *	0.01	0.169	0.01	0.03
Heart rate, min^−1^	75.7 ± 11.3	73.3 ± 11.0	76.9 ± 12.1	77.1 ± 10.3	0.03	0.170	0.01	0.049
Waist circumference, cm	89.7 ± 11.8	85.0 ± 10.6	91.1 ± 11.4 *	92.9 ± 12.0 *	<0.001	0.361	<0.001	<0.001
Weight, kg	66.3 ± 13.0	63.0 ± 11.8	68.0 ± 13.2 *	68.0 ± 13.6 *	0.01	0.213	0.001	0.02
BMI, kg/m^2^	26.5 ± 4.24	24.6 ± 3.88	26.7 ± 4.04 *	28.1 ± 4.06 *	<0.001	0.400	<0.001	<0.001
Body fat, %	34.3 ± 9.29	29.3 ± 8.68	34.9 ± 8.43 *	38.9 ± 8.13 *^,#^	<0.001	0.465	<0.001	<0.001
Laboratory Data								
Fasting glucose, mg/dL	113.7 ± 2.39	105.3 ± 37.6	116.4 ± 34.0	119.2 ± 40.6	0.01	0.237	<0.001	0.04
Total cholesterol, mg/dL	198.8 ± 42.9	202.1 ± 42.6	199.5 ± 39.7	194.9 ± 46.3	0.27	0.037	0.55	0.54
Triglyceride, mg/dL	115.0 ± 86.4	88.4 ± 55.3	132.6 ± 114.8 *	123.9 ± 72.4 *	0.01	0.227	<0.001	0.002
HDL, mg/dL	54.9 ± 19.3	61.1 ± 23.1	53.6 ± 18.2 *	49.9 ± 13.8 *	<0.001	0.229	<0.001	0.001
LDL, mg/dL	119.9 ± 35.8	120.6 ± 35.0	120.5 ± 35.5	118.5 ± 37.2	0.71	0.018	0.78	0.91
Uric acid, mg/dL	6.04 ± 1.51	5.58 ± 1.40	5.99 ± 1.41 *	6.45 ± 1.58 *	0.001	0.301	<0.001	0.003
e-GFR, mL/min/1.73 m^2^	79.1 ± 25.9	87.3 ± 23.0	81.8 ± 22.4	68.2 ± 28.1 *^,#^	<0.001	0.342	<0.001	<0.001
Biomarkers								
hs-CRP (median, 25th–75th), mg/L	0.22 ± 0.24	0.17 ± 0.19	0.21 ± 0.26	0.27 ± 0.26 *	0.01	0.255	<0.001	0.03
BNP (median, 25th–75th), pg/mL	62.2 ± 125.0	34.7 ± 80.0	55.5 ± 99.0	95.4 ± 169.8 *	0.002	0.207	0.001	0.01
Galectin-3, ng/mL	2.74 ± 2.36	2.16 ± 1.96	2.74 ± 2.16	3.32 ± 2.77 *	0.001	0.248	<0.001	0.001
PIIINP, ng/mL	0.98 ± 0.39	0.84 ± 0.26	0.98 ± 0.36 *	1.13 ± 0.46 *^,#^	<0.001	0.363	<0.001	<0.001
A-FABP, ng/mL	26.1 ± 21.4	17.4 ± 8.31	25.4 ± 14.7 *	35.7 ± 30.4 *^,#^	<0.001	0.392	<0.001	<0.001
Medical Histories								
Hypertension, *n* (%)	179 (71%)	47 (56.0%)	59 (70.2%)	73 (86.9%)	<0.001	—	—	<0.001
Diabetes, *n* (%)	75 (29.8%)	12 (14.3%)	25 (29.8%)	38 (45.2%)	<0.001	—	—	<0.001
Cardiovascular diseases, *n* (%)	34 (13.5%)	6 (7.1%)	10 (11.9%)	18 (21.4%)	0.01	—	—	0.02
Heart failure, *n* (%)	51 (20.2%)	5 (6.0%)	14 (16.7%)	32 (38.1%)	<0.001	—	—	<0.001
Metabolic score (median, 25th–75th)	3 (2–4)	2 (1–4)	3 (2–4) *	4 (3–5) *^,#^	<0.001	—	—	<0.001
Cardiac Structure and Function								
IVS, mm	9.20 ± 1.46	8.84 ± 1.29	9.15 ± 1.33	9.60 ± 1.65 *	<0.001	0.292	<0.001	<0.001
LVPW, mm	9.19 ± 1.29	8.81 ± 1.12	9.33 ± 1.38 *	9.42 ± 1.30 *	0.002	0.302	0	0.004
LVIDd, mm	46.3 ± 3.93	46.3 ± 4.20	46.6 ± 3.89	45.9 ± 3.71	0.56	0.001	0.98	0.55
LV mass, g	144.6 ± 37.0	137.3 ± 34.4	147.5 ± 37.0	149.0 ± 38.8	0.04	0.214	0.001	0.08
LV mass index, gm/m^2^	79.3 ± 18.8	76.9 ± 17.0	79.9 ± 20.6	81.2 ± 18.7	0.14	0.145	0.02	0.32
Stroke volume, mL	66.5 ± 12.4	67.0 ± 13.5	65.9 ± 11.4	66.5 ± 12.3	0.78	0.015	0.81	0.85
LVEF, %	67.1 ± 6.43	67.3 ± 6.20	65.8 ± 6.73 *	68.2 ± 6.19 *	0.38	0.040	0.52	0.05
LVH, *n* (%)	27 (10.7%)	6 (7.1%)	9 (10.7%)	12 (14.3%)	0.17	0.094	0.14	0.35
E/A ratio	0.92 ± 0.36	1.02 ± 0.394	0.89 ± 0.31	0.84 ± 0.34 *	0.001	0.281	<0.001	0.01
TDI-e’ (average), cm/s	7.71 ± 1.92	8.55 ± 1.98	7.76 ± 1.78 *	6.82 ± 1.60 *^,#^	<0.001	0.441	<0.001	<0.001
E/e’ (average)	9.84 ± 3.60	8.15 ± 2.71	9.90 ± 3.40 *	11.5 ± 3.85 *^,#^	<0.001	0.371	<0.001	<0.001
LV SRe, s^−1^	1.08 ± 0.30	1.23 ± 0.31	1.07 ± 0.29 *	0.96 ± 0.26 *^#^	<0.001	0.447	<0.001	<0.001
LV SRa, s^−1^	1.19 ± 0.24	1.17 ± 0.25	1.21 ± 0.22	1.19 ± 0.25	0.61	0.062	0.33	0.63
TDI-s’ (average), cm/s	7.62 ± 1.45	7.99 ± 1.42	7.80 ± 1.55	7.07 ± 1.20 *^,#^	<0.001	0.285	<0.001	<0.001
GCS, %	−20.6 ± 2.92	−20.7 ± 2.84	−20.9 ± 2.94	−20.3 ± 3.00	0.41	0.104	0.11	0.48
GLS, %	−19.5 ± 2.59	−20.5 ± 2.37	−19.4 ± 2.61 *	−18.5 ± 2.38 *	<0.001	0.408	<0.001	<0.001
LV SRs, s^−1^	−1.12 ± 0.15	−1.19 ± 0.155	−1.12 ± 0.13 *	−1.05 ± 0.13 *^,#^	<0.001	0.436	<0.001	<0.001

A, late diastolic filling velocity; BMI, body mass index; BNP, brain natriuretic peptide; CRP, C-reactive protein; DM, diabetes mellitus; EAT, epicardial adipose tissue; E/E’, relationship between maximal values of passive mitral inflow (E, PW-Doppler) and lateral early diastolic mitral annular velocities (E’, TDI); e-GFR, estimated glomerular filtration rate; FABP, fatty acid–binding protein; GCS, global circumferential strain; GLS, global longitudinal strain; HDL-c, high-density lipoprotein cholesterol; hs-CRP, high-sensitivity C-reactive protein; IVS, inter-ventricular septum; LA, left atrium; LDL-c, low-density lipoprotein cholesterol; LV, left ventricle/left ventricular; LVEDV, left ventricular end-diastolic volume; LVESV, left ventricular end- systolic volume; LVIDd, left ventricular internal diameter end diastole; LVPW, left ventricular posterior wall; PIIINP, procollagen type III N-terminal peptide; S’, peak systolic mitral annular velocity; SR, strain rate; SVi, stroke volume index; Tau (Ƭ), time constant of LV isovolumic pressure decline; TDI, tissue Doppler imaging. *p*-value < 0.05 for comparisons against * Q1 and ^#^ Q2.

**Table 2 diagnostics-11-00397-t002:** Association of EAT (as outcome measure) with baseline demographics on Pearson Correlation Test and Linear Regression Analysis

Variables	EAT (mm) (Multi-Variate Regression Model)			
	Adjusted Coefficient	*p* Value	Adjusted Coefficient	*p* Value
Age, years	0.03 (0.01, 0.05)	0.004	0.04 (0.01, 0.06)	0.001
Female sex, *n* (%)	0.5 (0.07, 0.93)	0.022	0.55 (0.12, 0.97)	0.012
Systolic blood pressure, mm Hg	0.01 (−0.0003, 0.02)	0.058	N/A	N/A
Diastolic blood pressure, mm Hg	N/A	N/A	0.02 (0.003, 0.03)	0.02
BMI, kg/m^2^	0.11 (0.06, 0.16)	<0.001	0.10 (0.06, 0.15)	<0.001
Fasting glucose, mg/dL	-	-	-	-
HDL-c, mg/dL	−0.012 (−0.023, −0.0003)	0.044	−0.012 (−0.02, −0.0006)	0.038
e-GFR, mL/min/1.73 m^2^	−0.012 (−0.02, −0.004)	0.003	−0.012 (−0.02, −0.004)	0.003
Hypertension, *n* (%)	-	-	-	-
Diabetes, *n* (%)	0.78 (0.35–1.22)	<0.001	0.81 (0.37–1.24)	<0.001
HFpEF, *n* (%)	0.81 (0.30–1.33)	0.002	0.83 (0.32–1.34)	0.002

CI: Confidence interval. Other abbreviations are as shown in Table 1.

**Table 3 diagnostics-11-00397-t003:** Multivariate cox regression models in predicting hospitalization for HF events.

Cox Regression Models	EAT (per 1 mm Increment)	
	HR (95% CI)	*p*-Value
Hospitalization for HF		
Crude Model	1.66 (1.45–1.91)	<0.001
Multivariate model	1.36 (1.14–1.64)	0.001
Composite HF/Death		
Crude Model	1.60 (1.41–1.83)	<0.001
Multivariate model	1.35 (1.14–1.60)	0.001

Multivariate model was adjusting for age, sex, BMI, blood pressure, DM, hypertension, cardiovascular disease, heart failure history, eGFR, and LV mass. BMI, body mass index; DM, diabetes mellitus; EAT, epicardial adipose tissue; eGFR, estimated glomerular filtration rate; LV, left ventricle/ventricular.

## Data Availability

Owing to local institutional regulation (which in this study was approved years ago and at that stage the authors did not apply for data spread or distribution out of the institution), together with the newly applied "Personal Information Protection Act" in Taiwan, the data will not be appropriate to be released in public place. The spread and data release will cause some concern from local ethical committee based on current institution regulations. Data are available from the "MacKay Memorial Hospital" Institutional Data Access / Ethics Committee for researchers who meet the criteria for access to confidential data. The contact information as follows: Mackay Memorial Hospital, Address: No. 92, Sec. 2, Zhongshan N. Rd., Taipei City 10449, Taiwan Tel: 02-25433535#3486~3488, Email: mmhirb82@gmail.com (Institutional Review Board).

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
