# Peer review of "Epicardial Adiposity in Relation to Metabolic Abnormality, Circulating Adipocyte FABP, and Preserved Ejection Fraction Heart Failure"

_diagnostics, 2021, doi:10.3390/diagnostics11030397_

Round 1

Reviewer 1 Report

The authors conducted a prospective study to the correlation between EAT size and cardiac function and remodeling using echocardiography. The authors further suggest that EAT size is associated with A-FABP level. EAT thickness was highest in participants with HFpEF and lowest in healthy controls. Importantly, they found that higher EAT correlated with the presence of cardiometabolic syndrome, diabetes, and renal insufficiency independent of BMI and waist circumference. A-FABP levels mediated the association between EAT and the development of HF, suggesting a metabolic link between EAT and HF, especially in patients with HFpEF.  The results in this elaborate study are convincing and important for the improvement of HFpEF and HF diagnosis.

I have a few comments regarding the manuscript:

  • Page 11: The authors declare “After a more detailed adjustment for baseline medical histories, only FABP was found to be the only mediator”. It's hard to believe that FABP is the only mediator associated with EAT and the risk of HF. It would be interesting to see in this  model if oxidative stress, TNFα, or TLR4 that have been shown to be associated with cardiac function and HF might also be associated with EAT size.
  • The authors should discuss the EAT characteristic compared to other adipose tissue and explain why EAT has more effect on cardiac function 
  • The authors might be willing to compare their results to abdominal fat size, to explore if their findings specific to EAT?
  • The specific contribution of diabetes to EAT size (independently of obesity) is not clear from this study. Is there?
  • Page 12: “A-FABP (also known as FABP4 or aP2) is a member of the intracellular FABP family predominantly expressed in mature adipocytes and has been shown to express higher levels in human EAT and central aortic vasculature in overweight/obese subjects or those with severe MetS [36] and associated with unfavorable LV remodeling in obese women [37].” This paragraph is more suitable in the introduction.
  • Assessment of epicardial fat by transthoracic echo is quite challenging and not accurate in obese subjects. A correlation of r=0.71 with CT is modest. While in large samples the differences may be significant, there is quite an overlap between the groups. This point needs to be addressed in LIMITATIONS.

Author Response

The authors conducted a prospective study to the correlation between EAT size and cardiac function and remodeling using echocardiography. The authors further suggest that EAT size is associated with A-FABP level. EAT thickness was highest in participants with HFpEF and lowest in healthy controls. Importantly, they found that higher EAT correlated with the presence of cardiometabolic syndrome, diabetes, and renal insufficiency independent of BMI and waist circumference. A-FABP levels mediated the association between EAT and the development of HF, suggesting a metabolic link between EAT and HF, especially in patients with HFpEF.  The results in this elaborate study are convincing and important for the improvement of HFpEF and HF diagnosis.

I have a few comments regarding the manuscript:

  • Page 11: The authors declare “After a more detailed adjustment for baseline medical histories, only FABP was found to be the only mediator”. It's hard to believe that FABP is the only mediator associated with EAT and the risk of HF. It would be interesting to see in this model if oxidative stress, TNFα, or TLR4 that have been shown to be associated with cardiac function and HF might also be associated with EAT size.

Response:

We thank the reviewer’s comment on this. Indeed, after detailed adjustment (including age, sex, bmi, Diabetes, smoker, HTN, Heart failure history, CVD), both hs-crp and FABP showed significance as potential mediators associated with EAT and the risk of HF (as the following figure); however, the p value for hs-CRP was marginal (p=0.05 for step 1 and 2; p=0.07 for explaining HF outcomes, as following results pasted), with FABP meet criteria as main mediator. If we just adjusted for age, sex and bmi, more biomarkers including BNP, PIIINP and FABP fulfilled criteria as mediators.

We therefore considered that FABP to be the mediator after further adjustment for baseline medical histories, however, this did not exclude other biomarkers as potential mediators when baseline medical histories were not included in models. Therefore, we believe that this result main partly reply on what clinical co-variates were put into the models. We did not have biomarkers of TNFα, or TLR4 and therefore they were not tested.

We hope this information may help to address this question further in a more detailed manner, and would consider to put hs-CRP as marginal mediator if the reviewer insists so.

  • The authors should discuss the EAT characteristic compared to other adipose tissue and explain why EAT has more effect on cardiac function 

Response:

Yes, we can since then added a short discussion, together with a citation, on this point accordingly (in page 10, line 336-342).

  • The authors might be willing to compare their results to abdominal fat size, to explore if their findings specific to EAT?

Response:

Yes, we completely agree on this point. However, our sample collection did not contain information about abdominal visceral fat and therefore we are not able to perform such statistical analysis. We had since then put this into our “Limitation section” accordingly.

  • The specific contribution of diabetes to EAT size (independently of obesity) is not clear from this study. Is there?

Response:

Yes, we thank the reviewer’s comment on this. We did multi-variate regression analysis in Table 2 and revealed diabetes, along with other clinical co-variates, was among the independent predictors for greater EAT. These descriptions have been well addressed in the following sentence (in page 5, line 227-231):

More advanced age, female sex, greater BMI, higher blood pressure, lower HDL-C, worsened renal function, and presence of diabetes, presence of HFpEF all were independent determinants for higher EAT (all p < 0.05) in multivariate regression models (with SBP and DBP were entered into models separately) (Table 2).

  • Page 12: “A-FABP (also known as FABP4 or aP2) is a member of the intracellular FABP family predominantly expressed in mature adipocytes and has been shown to express higher levels in human EAT and central aortic vasculature in overweight/obese subjects or those with severe MetS [36] and associated with unfavorable LV remodeling in obese women [37].” This paragraph is more suitable in the introduction.

Response:

Yes, we had since then moved this paragraph to Introduction section accordingly (in page 2, line 58-62).

  • Assessment of epicardial fat by transthoracic echo is quite challenging and not accurate in obese subjects. A correlation of r=0.71 with CT is modest. While in large samples the differences may be significant, there is quite an overlap between the groups. This point needs to be addressed in LIMITATIONS.

Response:

  • Yes, we had since then added this part as one of the study limitations accordingly.

Reviewer 2 Report

The authors have addressed an interesting topic. Using epicardial adipose tissue (EAT) they identified a risk factor for cardiovascular disease. As echocardiography is frequently performed, this provides additional information. The follow-up data are nice. The presentation is complete, however, this reviewer has additional comments:

during follow-up interventions were performed, for instance the start of medication, statins, ACE-i. Did the study population size allowed an interaction for the use of medication? In the same vein, could repeated measurements be performed?

Author Response

The authors have addressed an interesting topic. Using epicardial adipose tissue (EAT) they identified a risk factor for cardiovascular disease. As echocardiography is frequently performed, this provides additional information. The follow-up data are nice. The presentation is complete, however, this reviewer has additional comments:

during follow-up interventions were performed, for instance the start of medication, statins, ACE-i. Did the study population size allowed an interaction for the use of medication? In the same vein, could repeated measurements be performed?

Response:

Yes, we thank the reviewer’s comment on this point. These medications were used prior to the enrollment of our study, and therefore we treat them as binary data (Yes vs None for active use) to test the interaction analysis in current study, which will make the analysis more simple and doable.

We did test the associations of medication use on our main outcome measures of heart failure hospitalization or death. Among all study participants, there were 59.0% of ACEi/ARB use and 33.3% statin use. The interaction hazard ratio (HR) using adjusted Cox survival model showed 0.76 (95% CI: 0.57-1.09), P interaction: 0.11, for ACEi/ARB x EAT as interaction term; the hazard ratio (HR) using Cox survival model showed 0.85 (95% CI: 0.62-1.17), P interaction: 0.31. The data seemed not showing modifying effect between these medications sue and EAT burden on clinical endpoint.

Reviewer 3 Report

GENERAL COMMENT

This is a single-center prospective observational study aimed at evaluating the relationship between EAT as expression of visceral adiposity and some circulating biomarkers of inflammation and heart failure and its role in predicting adverse outcomes in patients with HFpEF.

The study is well conducted and methodologically comprehensive despite the relatively small size of the three study groups. Although with this and some other limitations, the authors report interesting results: EAT correlates linearly with the biomarkers studied and in particular the EAT burden (expressed as per 1-mm EAT increment) is an independent predictor of high A-FABP values; EAT burden is an independent predictor of HF and composite HF / death endpoint; the combined use of EAT and FABP after determining the optimal cut-off values using ROC curve analysis allowed patients to be stratified into 4 subgroups with increasing risk of HF events in the follow-up.

SPECIAL COMMENTS

Some improvements are recommended:

First, a thorough revision of the English language.

Secondly, the figures and tables are excessive and sometimes redundant, confusing rather than illustrative! To improve understanding, here are some suggestions:

  • table 1: keep (it is exhaustive)
  • figure 1: remove
  • Table 2: keep, but remove non-significant variables
  • figure 2: remove (part B: data is already included in table 1, only tertiles instead of quartiles; part A: models add no value and create confusion)
  • figure 3: keep but select only one model (simplify!)
  • figure 4: keep, is the key result of the study in my opinion
  • table 3: keep but select only one model (in my opinion model 3) and enter the single variables and their p-value as well as HR (95% CI) and p-value of the global model
  • figure 5: a little pretentious but undoubtedly interesting; keep

Author Response

GENERAL COMMENT

This is a single-center prospective observational study aimed at evaluating the relationship between EAT as expression of visceral adiposity and some circulating biomarkers of inflammation and heart failure and its role in predicting adverse outcomes in patients with HFpEF.

The study is well conducted and methodologically comprehensive despite the relatively small size of the three study groups. Although with this and some other limitations, the authors report interesting results: EAT correlates linearly with the biomarkers studied and in particular the EAT burden (expressed as per 1-mm EAT increment) is an independent predictor of high A-FABP values; EAT burden is an independent predictor of HF and composite HF / death endpoint; the combined use of EAT and FABP after determining the optimal cut-off values using ROC curve analysis allowed patients to be stratified into 4 subgroups with increasing risk of HF events in the follow-up.

SPECIAL COMMENTS

Some improvements are recommended:

First, a thorough revision of the English language.

Response:

Yes, we thank the reviewer’s comment on this. We had since then sent out for thorough language edition accordingly. Attached we will show the certification and proof of our English language edition (final page) during revision process.

Secondly, the figures and tables are excessive and sometimes redundant, confusing rather than illustrative! To improve understanding, here are some suggestions:

  • table 1: keep (it is exhaustive)
  • figure 1: remove

Response:

Yes, we had removed Figure 1 accordingly. Relevant information was supplemented in text accordingly.

  • Table 2: keep, but remove non-significant variables

Response:

Yes, we had removed those variables statistically non-significant accordingly.

  • figure 2: remove (part B: data is already included in table 1, only tertiles instead of quartiles; part A: models add no value and create confusion)

Response:

Yes, we had removed Figure 2 accordingly.

  • figure 3: keep but select only one model (simplify!)

Response:

Yes, we had removed 1 and kept only 1 model accordingly. Now figure 3 becomes figure 1 by sequence.

  • figure 4: keep, is the key result of the study in my opinion

Response:

Yes, Now figure 4 becomes figure 2 by sequence.

  • table 3: keep but select only one model (in my opinion model 3) and enter the single variables and their p-value as well as HR (95% CI) and p-value of the global model

Response:

Yes, we had dropped the other models accordingly.

  • figure 5: a little pretentious but undoubtedly interesting; keep